# Heterologous Hyaluronic Acid Production in *Kluyveromyces lactis*

**DOI:** 10.3390/microorganisms7090294

**Published:** 2019-08-28

**Authors:** Antonio M. V. Gomes, João H. C. M. Netto, Lucas S. Carvalho, Nádia S. Parachin

**Affiliations:** Grupo de Engenharia de Biocatalisadores, Departamento de Biologia Celular, Instituto de Ciências Biológicas, Universidade de Brasília (UnB), Campus Darcy Ribeiro, Bloco K. Brasilia, Federal District 70790-900, Brazil

**Keywords:** Hyaluronic Acid, Hyaluronic Acid Synthase, *Kluyveromyces lactis*, *Pasteurella multocida*, *Xenopus laevis*

## Abstract

Hyaluronic Acid (HA) is a biopolymer composed by the monomers Glucuronic Acid (GlcUA) and N-Acetyl Glucosamine (GlcNAc). It has a broad range of applications in the field of medicine, being marketed between USD 1000–5000/kg. Its primary sources include extraction of animal tissue and fermentation using pathogenic bacteria. However, in both cases, extensive purification protocols are required to prevent toxin contamination. In this study, aiming at creating a safe HA producing microorganism, the generally regarded as safe (GRAS) yeast *Kluyveroymyces lactis* is utilized. Initially, the *hasB* (UDP-Glucose dehydrogenase) gene from *Xenopus laevis* (xl*hasB*) is inserted. After that, four strains are constructed harboring different *hasA* (HA Synthase) genes, three of humans (hs*hasA1*, hs*hasA2*, and hs*hasA3*) and one with the bacteria *Pasteurella multocida* (pm*hasA*). Transcript values analysis confirms the presence of *hasA* genes only in three strains. HA production is verified by scanning electron microscopy in the strain containing the pmHAS isoform. The pmHAS strain is grown in a 1.3 l bioreactor operating in a batch mode, the maximum HA levels are 1.89 g/L with a molecular weight of 2.097 MDa. This is the first study that reports HA production in *K. lactis* and it has the highest HA titers reported among yeast.

## 1. Introduction

Hyaluronic acid (HA) is a glycosaminoglycan (GAG) composed by two molecules: the glucuronic acid (GlcUA) and *N*-acetyl glucosamine (GlcNAc) [1] (Figure 1). It is absent in insects and plants and occurs naturally in all vertebrates, also in algae and mollusks [2]. In the vast majority of fungi, HA is absent, but in some cases, as in the yeast *Cryptococcus neoformans* [3], this polymer is present. In vertebrates, HA is known for participating in cell migration, communication, adhesion, growth, and differentiation, as well as tissue repair and angiogenesis [4]. Is expected that the HA global market will reach approximately U$D 15.25 Billion by 2026 with a Compound Annual Growth Rate (CAGR) of 7.8% [5]. Its increased demand over the years is associated with its application in various fields related to aesthetics, pharmaceuticals, and medicine. Currently, several medical techniques use HA such as in orthopedic [6] and ophthalmologic procedures and for wrinkle reduction in anti-aging treatment [7].

The metabolic route to produce HA is highly conserved among different organisms (Figure 1). Both UDP-glucuronic acid (UDP-GlcUA) and UDP-N-acetyl glucosamine (UDP-GlcNAc) are synthesized from glycolytic precursors, the first from glucose-6-phosphate and the latter from fructose-6-phosphate (Figure 1). The last step during HA synthesis is catalyzed by the Hyaluronan Synthases or HA Synthases (HAS). HAS enzymes are coupled to cell membranes of organisms and are responsible for alternating the ligation of UDP-GlcUA and UDP-GlcNAc and exporting it to the extracellular environment (Figure 1). HAS enzymes are divided into two distinct classes: class I comprise all vertebrates, bacteria and a few other microorganisms and class II present only in the bacteria *Pasteurella multocida* [8,9,10,11,12,13]. The latter has differences in the transmembrane domains, structural organization, and mode of activity [11]. In class II enzyme, the elongation of HA is by the non-reducing end, different from all other HAS enzymes belonging to class I [13]. Among others, the class I of HAS comprises the three isoforms present in humans (*Homo sapiens*) that are encoded, respectively, by the genes hs*hasA1*, hs*hasA2*, and hs*hasA3* [12]. These three isoforms have different kinetic parameters, which result in an HA with different chain sizes [12].

Primarily, HA was obtained from rooster combs, bovine vitreous humor, and umbilical cord [14]. In animal tissues, HA is coupled with proteoglycans that need to be removed from the biopolymer to allow its commercialization. These processes are laborious and costly and, therefore, novel processes for HA production based on microbial fermentation were developed. Naturally, HA production has been reported in the *Streptococci* genus [15]. Only this group of bacteria and *Pasteurella multocida* can naturally synthesize HA. Usually, fermentations using *Streptococcus* cells results in titers ranging from 0.3 up to 4.6 g/L [16] with polymer sizes ranging from 1 to 4 Mega Daltons (MDa). However, because of its pathogenicity factors, the HA produced by *Streptococci* cells also requires a rigorous and costly purification process. 

Thus, the HA biosynthetic pathway has been transferred to other non-pathogenic microorganisms such as *Lactococcus lactis* [17], *Escherichia coli* [18], *Bacillus subtilis* [19,20,21], *Agrobacterium* sp. [22], *Corynebacterium glutamicum* [23], *Streptomyces albulus* [24], and plant cell cultures [25]. In yeasts, *Pichia pastoris (Komagataella *spp*.)* has been previously utilized as a platform for HA production [26]. Here, for the first time, the *Kluyveromyces lactis* metabolism was modified for enabling HA production. *K. lactis* has a status generally regarded as safe (GRAS) and is commonly utilized to produce dairy products. Moreover, unlike other yeasts, *K. lactis* is advantageous for HA production because: (I) its respiratory metabolism with absence of ethanol production; (II) its central metabolism with carbon flow diverted to Pentose Phosphate Pathway (PPP) [27] that produces precursors necessary for HA synthesis, (III) its availability of sequenced genome [28] and establishment in genetic manipulation protocols, (IV) it is a consolidated platform for heterologous protein production [29], and finally (V) during process optimization, it is possible to achieve high cell density in *K. lactis* using low-cost carbon sources such as lactose and whey, which also serve as inductors of strong promoters in this yeast. 

According to the genome annotation of the *K. lactis* NRRL Y-1140 strain, the biosynthetic pathway of HA is incomplete. The yeast possesses all precursors for the synthesis of UDP-GlcNAc (Figure 1) while the metabolic pathway for the UDP-GlcUA production is incomplete (Figure 1). Therefore, to overcome this barrier, two genes were inserted into the *K. lactis* genome to complete the HA synthesis: (i) the *hasB* (UDP-Glucose Dehydrogenase) gene from *Xenopus laevis* (xl*hasB*) and (ii) the *hasA* (HA Synthase). In all constructed strains, the same xl*hasB* gene was utilized in combination with different versions of the *hasA* encoding gene. 

Up to now, heterologous microorganisms that synthesize HA generally uses the class I of HAS enzymes [2]. Therefore, here for the first time HAS of class I and II were used to compare the HA production. Four distinct recombinant strains of *K. lactis* were constructed (Figure 2) containing each the three humans (*Homo sapiens*) Class I *hasA* genes (hs*hasA1*, hs*hasA2,* and hs*hasA3*) and the Class II *hasA* gene from *Pasteurella multocida* (pm*hasA*). Both the *hasA* genes from *Homo sapiens* [12,30,31,32,33] and *P multocida* [34] were previously characterized and for this reason they were chosen for insertion in *K. lactis*. In addition, *Xenopus laevis* is a standard organism used in the study of HA synthesis [35], and the genes (*hasA* and *hasB*) of this organism have also been previously characterized.

## 2. Materials and Methods

### 2.1. Microorganisms and Plasmids

All strains and plasmids used in this study are described in Table 1. All plasmids maps and sequences used during this study are described in the Appendix A. Plasmid replication was done in *E. coli* strain XL-10 Gold using the heat shock protocol as previously described [36]. Transformants cells were selected in LB plates (yeast extract 5 g/L, tryptone 10 g/L, NaCl 10 g/L and agar 20 g/L in pH 7.0) containing ampicillin in a final concentration of 100 µg/mL.

### 2.2. Plasmids Construction

The overall process for plasmid construction is summarized in Figure 2. All heterologous genes used in this study were previously optimized for expression in *K. lactis* and delivered in the pBSK vector (Table 1).

Initially, the xl*hasB* gene was removed from the pBSK-HASB and cloned into the plasmid p424GPD (Figure 2). After that, the construction of the *K. lactis* expression vectors was initiated by amplifying the *hasB* expression cassette containing the GPD promoter and CYC1 terminator and inserted into pKlac2 (New England Biolabs Inc, Ipswich, MA, USA). The resulting plasmid was named pKlac2-B (Table 1). Next, for the insertion of *hasA* genes, all pBSK-HASAP, pBSK-HASA1, pBSK-HASA2, and pBSK-HASA3 plasmids were treated with *Hind*III and *Stu*I. The resulting fragments were ligated into the pKlac2-B previously treated with the same restriction enzymes. The four resulting plasmids, pKlac2-BP, pKlac2-B1, pKlac2-B2, and pKlac2-B3 (Table 1 and Figure 2), containing both xl*hasB* and *hasA* genes were sequenced before yeast transformation. All primers utilized in this study are summarized in Appendix A.

### 2.3. Nucleotide Sequence Accession Numbers

The codon-optimized sequences of a *hasB* gene from *Xenopus laevis, hasA* gene from *Pasteurella multocida* and the three *hasA* genes from *Homo sapiens* were deposited in GenBank under accession numbers MH728986, MH728990, MH728987, MH728988, and MH728989, respectively.

### 2.4. Transformation of Kluyveromyces lactis Cells

*K. lactis* strain GG799 cells were transformed using linearized plasmids with *SacII*. After linearization with *SacII*, the plasmids constructed from pKlac2 released a cassette for integration into the LAC4 promoter of *K. lactis* (Figure 2). Yeast transformation was done by using the “*K. lactis* Protein Expression Kit” from NEB (New England Biolabs) following the manufacturer’s recommendations. Transformants were selected in YCB acetamide agar (1.17% YCB medium, acetamide 5 mM, sodium phosphate buffer 30mM, and agar 2%). The plates were incubated at 30 °C for 3–4 days.

Gene integration was confirmed by colony PCR using the primers listed in Appendix A. Genomic DNA of strains was extracted using the GeneJET Genomic DNA Purification Kit (Thermo Fisher Scientific, MA, EUA) according to the manufacturer’s protocol. 

### 2.5. Growth of K. lactis Strains BAP, BA1, BA2, and BA3 in Shake Flask

All strains of *K. lactis* constructed in this study, in addition to the wild-type, GG799 strain, were grown in 250 mL shake flasks during 24 h containing 100 mL of a modified YPD medium as previously described [37]. Briefly the medium was composed by 7.5 g/L yeast extract, 10 g/L peptone, 40 g/L glucose, 2.5 g/L K_2_HPO_4_, 0.9 g/L MgSO_4_ 7 H_2_O, 5 g/L NaCl, 0.4 g/L glutamine and 0.6 g/L glutamate. The cells were grown at 30 °C and 200 rpm. Monitoring of cell growth in shake flasks was carried every 2 h with the reading of the cell density in a SpectraMax M2 microplate reader (Molecular Devices^®^) in OD_600_.

### 2.6. Detection of the hasB, hasA1, hasA2, hasA3 and hasAP Gene Transcripts

As described before, all strains constructed in this study were grown in shake flask until early exponential phase (OD_600_ = ~1.0). At this time, 2 mL of cells were collected and subjected to total RNA extraction using TRIzol reagent (Invitrogen, Carlsbad, EUA) according to the manufacturer’s recommendations. Only samples containing OD_260/280_ > 1.8 were utilized for transcript values analysis. 

For checking RNA integrity, all samples were analyzed on a bleach agarose gel as previously described [38] (Appendix A). After selecting the RNAs with a reasonable degree of purity and integrity, about 1.0 ug of total RNA was used as a template for reverse transcription using SuperScript^®^ III Reverse Transcriptase kit (Thermo Fisher Scientific) according to the manufacturer’s recommendations. The resulting cDNA from the wild-type strain GG799 and BAP, BA1, BA2 and BA3 strains was used as template in real-time PCR reaction using the following conditions: initial denaturation for 10 min at 95 °C; 40 cycles of 95 °C for 15 s-optimum temperature of each primer for 30 s-72 °C for 30 s, and 72 °C for 60 s. Each reaction of 20 µL was constructed using SYBR^®^ Green PCR Master Mix (Thermo Fisher Scientific) according to the manufacturer’s recommendations. The melting curve of all transcripts obtained in this study was used to visualize the correct specificity of each pair of primers (Appendix A).

### 2.7. Gene Copy Number Quantification

Gene copy number of pm*hasA* and xl*hasB* was confirmed only in BAP strain by qPCR as previously described [39]. The Actin gene was used as endogenous control because it is present in only one copy according to the online genome database of *K. lactis*. 

After genomic DNA extraction and purification, 100 ng of it was distributed in qPCR reactions containing dilutions of 100, 10, 1, 0.1 and 0.01 ng/reaction in a 20-μL mixture. The values of C_T_ obtained were used to construct an efficiency curve of each primer utilized. Only the reactions containing values of efficiency amplification between 90% and 110% were used for calculation of fold-change. 

Quantitative PCR (qPCR) was performed using SYBR^®^ Green PCR Master Mix (Thermo Fisher Scientific) according to the manufacturer’s recommendations with a StepOne^TM^ Real-Time PCR System (Applied Biosystems, CA, EUA).

### 2.8. Microscopic Analysis of Recombinant K. lactis Strains

Cell images were obtained on a field emission scanning electron microscope (Jeol JSM-7000F) with the assistance of a metalized (Emitec K550) and critical drying point (Emitec K850). 1 mL of cells were collected after 24 h of growth in a shake flask, as described in Section 2.5. The cells were prepared for analysis, as previously described [40]. 

### 2.9. Batch Fermentations of K. lactis Strains

Batch fermentations of the Wild-type (GG799) and BAP strains were performed in New Brunswick BioFlo^®^ 115 bioreactors (Eppendorf AG, Hamburg, Germany) with a capacity of 1.3 L. Pre-inoculums of both strains were grown in 50 mL of Yeast Nitrogen Base (YNB) containing ammonium sulfate and amino acids (Sigma-Aldrich -St. Louis, MO, USA) in a flask with a capacity of 0.5 L. The medium was prepared according to the manufacturer’s recommendations using glucose (40 g/L) as a carbon source. The strains grew for approximately 24 h at 30 °C and 200 rpm. This pre-culture was then used to inoculate 1 L of the same YNB medium in the fermenter with an initial OD_600_ = ~0.1. The fermentation was conducted at 30 °C, and 200 rpm in 1 L. During all fermentations, the pH of the medium was maintained in 6.0 with the automatic injection of NaOH 2 M from a feed base bottle, and the Dissolved Oxygen (DO) in 30% kept in cascade using stirring and air flow at 2 vvm. All batch fermentations were performed in biological triplicate.

### 2.10. Detection and Quantification of the Substrate and Cellular Products

Glucose consumption and production of acetate, ethanol, lactate, and glycerol were quantified using High-Performance Liquid Chromatography (HPLC) (Shimadzu, Kyoto, Japan) equipped with refractive index and UV (210-nm) detectors. The chromatography was performed using a Shim-pack SCR-101H (Shimadzu) (300 mm × 7.9 mm id) column equilibrated at 60 °C with 5 mM H_2_SO_4_ as the mobile phase at a flow rate of 0.6 mL/min. The run was 59 min long. Cell growth was monitored by withdrawing 1 mL aliquots of medium fermentation every 2 h for reading at 600 nm using a SpectraMax M2 microplate reader (Molecular Devices^®^). 

### 2.11. HA Quantification

For HA quantification, the culture broth fermentation of all strains and a wild-type strain control was diluted with 1 volume of 0.1% SDS for uncoupling the HA capsule surrounding the cell wall. After 10 min at room temperature, the cells were centrifuged 6000× *g* 4 °C. After the supernatant was filtered with 0.20 µm filter, then the HA was purified by washing the medium twice with 3–4 volumes of 100% ethanol. The HA pellet formed was resuspended in 50 mL of deionized water, and the Carbazol method was used for HA quantification as previously described [41]. A standard curve of HA concentrations ranging from 0.1 to 0.8 g/L was made using HA 99% Sigma-Aldrich (St. Louis, MO, USA). The samples were diluted 50× and 100× in deionized water before reading. 

### 2.12. HA Molecular Weight Determination

The molar mass of HA was analyzed by performing Aqueous Gel Permeation Chromatography (GPC). The samples of HA were analyzed with a concentration of 10 mg/mL. As a solvent, a 0.1 M NaNO_3_ solution with the flow rate of 0.5 mL/min was used in a Column System: 2SB-807 HQ, 2SB-806M HQ. The Injection Volume was 100 μL with Refractive Index Detectors and Poly (Ethylene Oxide) standards (PEO). The “hyaluronic acid sodium salt from *Streptococcus equi*” utilized as a standard in GPC analysis was obtained from Sigma Aldrich (St. Louis, MO, USA) and contained ~1.5 to 1.8 MDa according to the manufacturer information.

## 3. Results

### 3.1. Growth Rate Determination and HA Detection for BAP, BA1, BA2, and BA3 in Shake Flask

All strains constructed in this study were named concerning the versions of *hasA* genes as listed in Table 1. The strains containing the genes pm*hasA*, hs*hasA1*, hs*hasA2* and hs*hasA3* were respectively named BAP, BA1, BA2, and BA3. All strains were grown on shake flask, and the growth rate on glucose was calculated. The BAP strain had a more extended lag phase in comparison to all constructed strains and the wild type (Appendix A). All strains, including the wild type, showed statistically the same growth rate (Table 2). Finally, by the end of the exponential growth phase, the final biomass of the BAP strain was about half of that obtained by the other four strains (Appendix A). 

### 3.2. Transcript Values of HAS Genes

The analysis of transcripts values of all *hasA* and *hasB* genes were analyzed in all constructed strains. The actin transcript values were normalized to 100% to calculate the transcript values of the other genes. As it can be seen in Appendix A, the gene xl*hasB* was detected in all recombinant strains but at least with half of the transcript level when compared to actin. The *hasA* gene was detected only in strains BAP, BA1 and BA3 with about the same transcription levels but with about 60% of the transcript level for the actin gene. The number of copies of pm*hasA* and xl*hasB* genes was quantified only in the HA producing strain (BAP). Each gene presented three copies (Appendix A). 

### 3.3. Microscopic Analysis of GG799, BAP, BA1, BA2, and BA3 Strains

All constructed strains were analyzed by Electron Microscopy to investigate the presence of an HA capsule. As shown in Figure 3, the strains GG799 (Figure 3a), BA1 (Figure 3b), BA2 (Figure 3c), and BA3 (Figure 3d) did not produce any capsule. Nevertheless, the BAP strain was the only one that had a capsule around its cells corroborating HA production (Figure 3e,f). Indeed, in BAP, HA production could be detected already at 6-h of fermentation (Figure 3e).

### 3.4. Batch Fermentation and Co-Product Formation

The BAP and the wild-type GG799 strains were grown in a 1.3 L bioreactor to determine what was the possible changes in the co-product distribution resulted by the genetic modifications introduced for HA synthesis. As it can be seen in Table 3, the µ_Max_ values of BAP and GG799 are nearly the same. Nevertheless, the biomass yield was 19.5% lower in the BAP strain compared to the wild-type strain, corroborating a shift of carbon flux from cell growth to HA synthesis (Table 3). All these results were also observed in flasks’ growth (Table 2).

Concerning the other metabolic products, glycerol formation was reduced almost seven times while ethanol increased nearly eight times in the BAP strain. Lactate was detected only in the wild-type strain at negligible values, whereas acetate was only produced in BAP strain also at negligible levels (Table 3). The HA concentration, according with Carbazole method quantification, resulted in 1.89 g/L.

### 3.5. HA Molecular Weight Determination

GPC analyzes resulted in 3 elution peaks at the retention volumes of 27.88 mL, 37.89 mL and 44.87 mL, (Figure 4). The first (27.88 mL) and the later (44.87 mL) peaks represents compounds with molecular weight of 2.097 MDa and 192 Da, respectively. Considering that each disaccharide of HA contains ~400 Da [14], the latter elutes (37.89 and 44.87) correspond to impurities contained in the samples. The GPC analysis of the standard HA (containing ~1.5 MDa) obtained commercially and utilized for comparation is represented in Appendix A. The 2 peaks eluted in 32.75 mL and 46.13 mL in GPC analysis of HA standard represent populations with molecular weight of approximately 1.3 MDa and 126 Da, respectively (Appendix A).

## 4. Discussion

Except for *Cryptococcus neoformans* [42], yeasts are not able to produce Hyaluronic Acid. Up to now only the yeasts *S. cerevisiae* [43] and *P. pastoris* [26] (*Komagataella phaffii*) were previously genetically modified for HA production. In *P. pastoris*, for example, 5 genes (*hasA*, *hasB*, *hasC*, *hasD*, and *hasE*) were inserted. Similarly, to *P pastoris*, several other studies of HA production in genetic engineered modified microorganisms overexpress the *hasC*, *hasD* and *hasE* genes in addition to *hasA* and *hasB*. All these genes are generally chosen because in some AH producing bacteria, such as *Streptococcus zooepidemicus* subsp. *equi*, there is a dedicated operon for the synthesis of HA containing the genes *hasA*, *hasB*, *hasC*, *hasD*, and *hasE* [44]. Here HA production could be confirmed only by the addition of *hasA* and *hasB*. 

Up to now, the microorganisms having the same genetic modifications introduced in this study (only the addition of *hasB* and *hasA* genes) are summarized in Table 4. 

Most strains presented in Table 4 use the *hasA* gene from *Streptococcus equi subsp. zooepidemicus* while here, the class II *hasA* gene, from *Pasteurella multocida* was successfully inserted for the first time in *K. lactis* and resulted in maximum HA production of 1.89 g/L. HA production by BAP strain was also confirmed by electron microscopic analysis similarly to previous studies [52]. 

Among studies with yeasts, only *P. pastoris* is listed in Table 4. Differently, from *P. pastoris*, the Lac4 promoter of *K. lactis* is not repressed by glucose or other carbon sources. The insertion of *hasA* and *hasB* genes in *P. pastoris* resulted in an HA production of 200 mg/L, value approximately 9.5 times lower than the obtained here using *K. lactis* with the same genetic modifications. Furthermore, the highest HA concentration in *P. pastoris* is ~10% lower than the titers reached by *K. lactis.*


On the contrary, no HA could be detected in the recombinant *K. lactis* strains where three different *Homo sapiens hasA* genes were introduced even though the transcripts of *hasA* genes could be detected in BA1 and BA3. The human *hasA* genes have been previously characterized [12] whereas it has been suggested that once translated, these enzymes should pass to an unknown regulatory process before coupling it to the cell membrane. Furthermore, the mammalian HAS enzymes have eight domains which six are transmembrane ones, and two are membrane associated [52] which makes its insertion into the yeast membrane challenging. Another reason for the lack of activity on the recombinant strains containing the human *hasA* genes could be post-translation regulation. It has been previously shown that *hasA2*, for example, can be strongly inhibited by AMPK by phosphorylation of a threonine residue in this enzyme [53]. Furthermore, other molecules such as EGF, FGF2, FGF, Forskolin, IGF, IL-1β, PDGF, Progesterone, Prostaglandin, TGF-β, Estradiol, 4-MU, TGF-β1, and TGF-β1 have been previously described to control human HAS activities [54]. Since none of them are present in *K. lactis* this could justify the absence of activity of these enzymes. In addition, various studies suggest that human HAS enzymes can undergo a wide variety of post-translational modifications that can regulate the enzyme activity [52]. One of them is the glycosylation pattern, where O-GlcNAcylation and mono-ubiquitination were reported to influence HAS activities. Since the trend of mammalian glycosylation modifications is different from one found in yeasts, this may also have caused the absence of human HAS activities *K. lactis*. Altogether it can be said that although transcripts of hs*hasA1* and hs*hasA3* genes were detected in strains BA1 and BA3 respectively, a translation failure or more likely incorrect coupling of HAS enzymes into the yeast membrane may have prevented the enzyme activity. 

Among the 17 strains listed in Table 4, only three [19,24,53] showed HA titers higher than those obtained in this study. However, in one of them [24], the HA production of 5.1 g/L with *Streptomyces albulus* has been achieved maintaining the glucose concentration at 5% in a fed-batch mode. The other study with *Corynebacterium glutamicum* [53] achieved an HA production (5.1 g/L) 2.85 times higher than *K. lactis* with the same initial substrate concentration. However, in *Corynebacterium glutamicum,* a strategy of gene induction with the pTAC promoter (IPTG induction after initial growth) was utilized, while in *K. lactis* HA was produced during the entire growth phase. Finally, in *Bacillus subtilis* [19], HA production was ~8% higher, however, a higher concentration of substrate was used in relation to *K. lactis*. 

As shown in Table 4, the Carbazole method [41] for HA quantification was used in 13 of the 17 studies. The carbazole protocol is based in the quantification of GlcUA present in a sample. For this, the HA must be previously hydrolyzed from the HA chains with a strong acid (H_2_SO_4_) treatment at high temperature (100 °C).The carbazole method is accurate but is strongly influenced by residual concentrations of salts and carbon sources present in the medium [55] that are co-purified together with HA. The impurities present in the sample react with H_2_SO_4_ during hydrolysis changing the Carbazole assay color and giving overestimated results. In this study, the HA purified from growth in shake flasks could not be properly determined due to impurities of the rich medium (YPD) (data not shown). Therefore, HA titer is only reported from culture supernatant using defined medium. Finally, the protocol for Carbazole method utilized for HA quantification in *K. lactis* is identical to protocol utilized in most of the 13 studies shown in Table 4, and the Calibration Curve is presented in Appendix A. 

Regarding HA Molecular Weight, only a minority of strains (Table 4) achieved similar values. Usually, HA applications depend on its molecular weight for example in mammals, high molecular HA weight (>1 MDa) have a role in maintaining cell integrity [56], while low molecular weight chains (<10^4^ Da) are used as receptors and signaling agents during cell communication [54]. In this study, the HA produced by *K. lactis* in defined medium reached 2.09 MDa by using GPC analysis. Although GPC analysis may result in an overestimation of up to two times [57], any values above 1 MDa are considered high molecular weight HA [2]. In addition, it is important to emphasize that the GPC analysis of the standard HA utilized in this study showed a value of 1.3 MDa while the supplier information determined it as 1.5 MDa (Appendix A) which shows that the error of the GPC assay is not higher than 16%. 

The concentration of HA above 2 g/L simultaneously with a high molecular weight (>1 MDa) is rare since it has been reported that there is an inverse relation regarding HA titers and molecular weight [52]. Besides, the ratio among Glucuronic Acid and N-Acetyl Glucosamine have also shown to influence HA molecular weight and concentration. For example, in *S. zooepidemicus*, the overexpression of genes from the UDP-Glucuronic Acid synthesis pathway decreases the molecular weight of the HA and increases its titers, whereas the overexpression of the UDP-N-Acetyl Glucosamine has the opposite effect [58]. On the contrary, in *Bacillus subtillis*, when the three genes, *hasA*, *hasB*, and *hasC* involved in the synthesis of UDP-Glucuronic Acid were super expressed, the resulting HA had an increased molecular weight when compared to the recombinant strain where the levels of Glucuronic Acid and N-Acetyl Glucosamine were balanced by the overexpression of *hasD* [20]. Therefore, the influence of precursor concentration and activities of critical enzymes is suggested to be cell-dependent and yet needs to be further investigated in *K. lactis*. 

The metabolic pathway for HA synthesis highlights how energetically costly the polymer synthesis is for the host cell (Figure 1). The production of one monomer of HA requires 3 ATP, 2 UTP, 2 NAD^+^, 1 Acetyl-CoA, and one glutamine. Indeed, availability of ATP has been shown to increase HA production. For example, the utilization of *Streptomyces albulus*, a bacterium able to synthesize ATP molecules at a higher level when compared to other bacteria, as host of HA production was able to produce up to 6.2 g/L of HA [24], one of the highest HA titers reported up to now. Furthermore, HA synthesis generates an accumulation of NADH. Therefore, to keep the cell redox balance during HA synthesis, it is necessary to activate metabolic pathways that reoxidize NADH. *K. lactis* is a Crabtree-negative yeast and does not produce ethanol under aerobic conditions. This was corroborated here in aerobic batch fermentation (Table 3). Nevertheless, the introduction of xl*hasB* and pm*hasA* genes resulted in an increase of approximately seven times in ethanol yield in the BAP strain (Table 3) probably to favor recycling of NAD^+^. In contrast, decreased glycerol production in the modified BAP strain was not expected since glycerol production also reoxidizes NADH. Nevertheless, the fructose 6-phosphate precursor is both used in HA and glycerol synthesis (Figure 1). Thus, this shift may have caused the reduction of glycerol formation in the cell. Finally, in both shake flasks and bioreactor fermentations, HA synthesis did not affect the yeast growth rate (µ_MAX_) but decreased the biomass yield from glucose in comparison with the wild type.

## 5. Conclusions

The addition of human *hasA* genes in *K. lactis* genome does not result in the synthesis of Hyaluronic Acid. However, the addition of the *hasA* gene from *Pasteurella multocida* in combination with the *hasB* gene from *Xenopus leavis* enabled HA production at the concentration of 1.89 g/L with a molecular weight of 2.09 MDa. This HA production does not affect the yeast growth rate compared to the wild-type strain, but changes the final yield of biomass, glycerol, and ethanol. Altogether our results are the proof of principle for HA production in *K. lactis* having competitive titers when compared to other microorganisms. It can be used as a basis for the development of an industrial bioprocess for HA production using *K. lactis* as the biocatalyst.

## Figures and Tables

**Figure 1 microorganisms-07-00294-f001:**
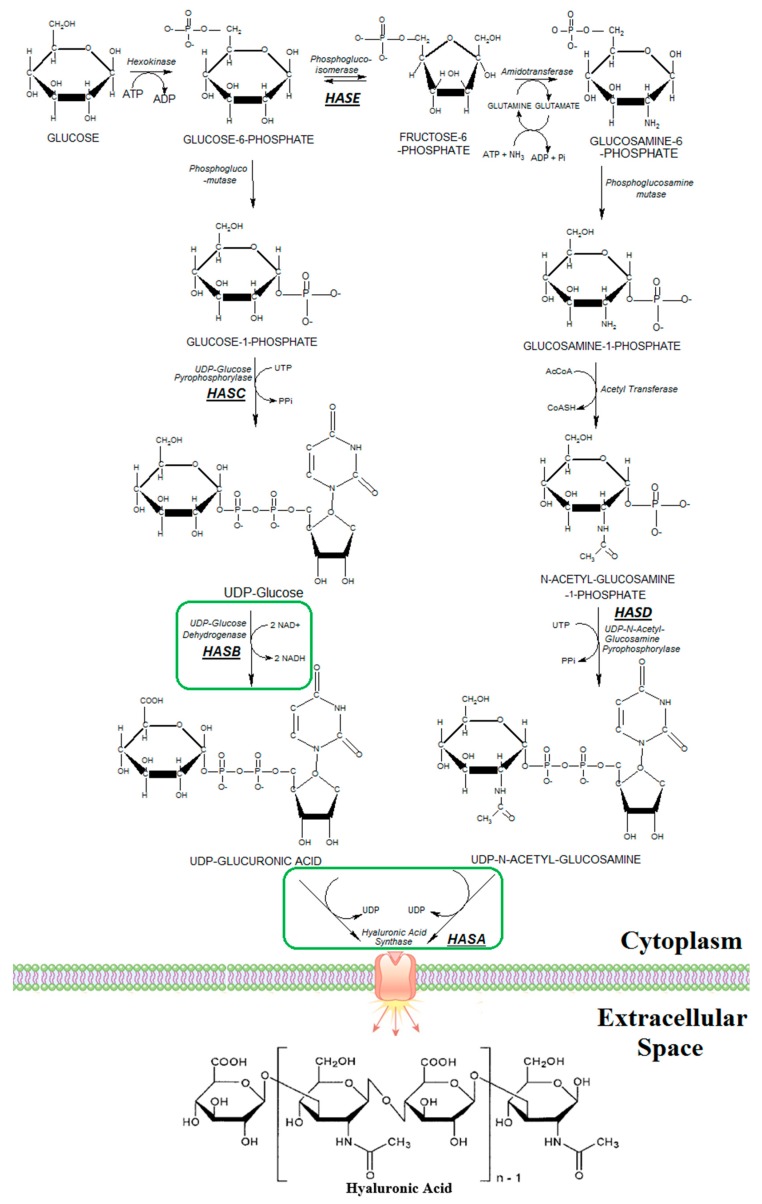
The metabolic pathway for Hyaluronic Acid production. In green, the enzymes that are not present in the metabolism of *K. lactis* but are necessary for the synthesis of Hyaluronic Acid.

**Figure 2 microorganisms-07-00294-f002:**
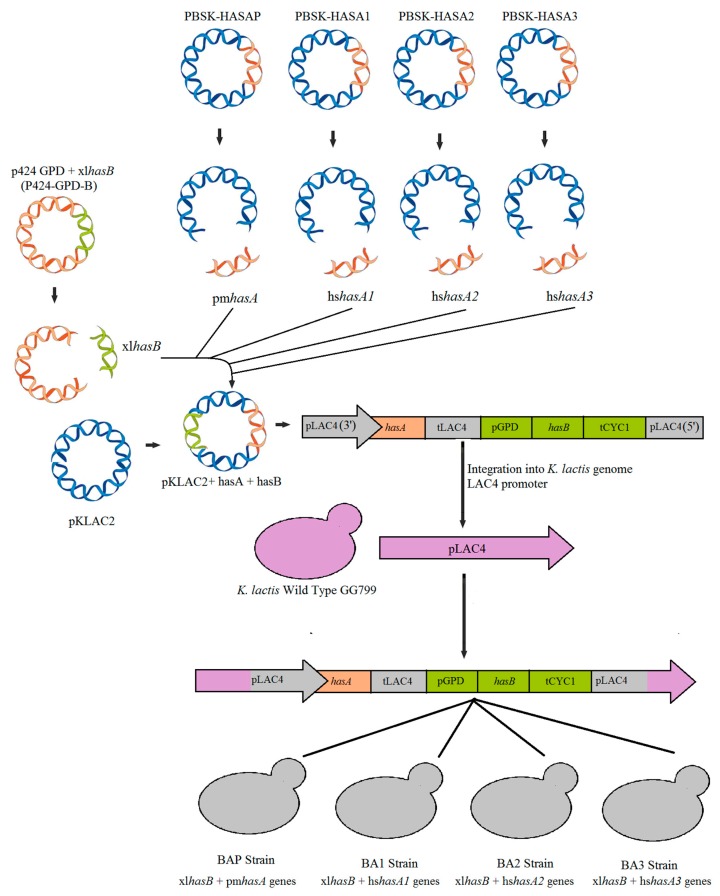
Overall construction strategy for the four strains generated in this study. The *hasB* gene from *Xenopus laevis* (xl*hasB*) and the different versions of *hasA* genes from *Homo sapiens* (hs*hasA1*, hs*hasA2*, and hs*hasA3*) and *Pasteurella multocida* (pm*hasA*) were inserted into the integrative plasmid pKlac2 for recombination in the LAC4 promoter of the *K. lactis* genome.

**Figure 3 microorganisms-07-00294-f003:**
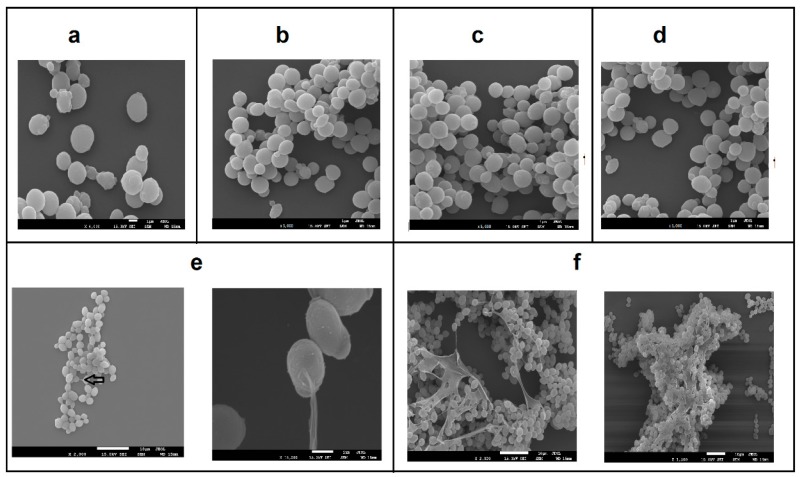
Scanning electron microscopy (SEM) analysis of the wild-type *K. lactis* GG799 strain and all strains constructed in this study. (**a**) GG799 strain (6000-fold increase) after 24 h of flask cultivation; (**b**) BA1 strain (5000-fold increase) after 24 h of flask cultivation; (**c**) BA2 strain (5000-fold increase) after 24 h of flask cultivation; (**d**) BA3 strain (5000-fold increase) after 24 h of flask cultivation; (**e**) BAP strain (5000-fold and 15000-fold increase) after 6 h of flask cultivation and (**f**) BAP strain after 24 h of flask cultivation in 5000-fold increase.

**Figure 4 microorganisms-07-00294-f004:**
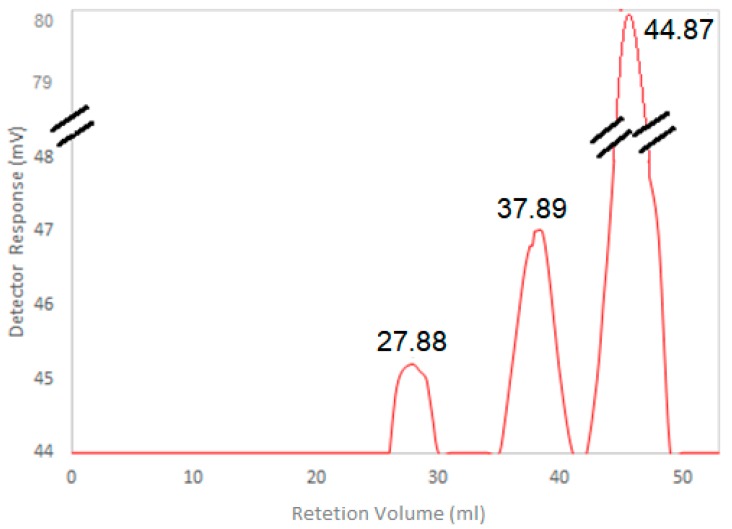
Gel Permeation Chromatography (GPC) analysis of the Hyaluronic Acid (HA) purified from BAP strain after growth in bioreactor.

**Table 1 microorganisms-07-00294-t001:** Plasmids and strains used in this study.

**Plasmids**	**Description**	**Reference**
PBSK-HASB	The synthetic plasmid containing xl*hasB* optimized	This study
PBSK-HASA1	The synthetic plasmid containing hs*hasA1* optimized	This study
PBSK-HASA2	The synthetic plasmid containing hs*hasA2* optimized	This study
PBSK-HASA3	The synthetic plasmid containing hs*hasA3* optimized	This study
PBSK-HASAP	The synthetic plasmid containing pm*hasA* optimized	This study
P424-GPD	Commercial Plasmid containing GPD Promoter	ATCC^®^
P424-GPD-B	P424-GPD + xl*hasB* gene	This study
PKLAC2	Commercial Plasmid for *K. lactis* protein expression	New England Biolabs
PKLAC2-B	pKlac + xl*hasB* gene	This study
PKLAC2-BP	pKlac + xl*hasB* gene + pm*hasA* gene	This study
PKLAC2-B1	pKlac + xl*hasB* gene + hs*hasA1* gene	This study
PKLAC2-B2	pKlac + xl*hasB* gene + hs*hasA2* gene	This study
PKLAC2-B3	pKlac + xl*hasB* gene + hs*hasA3* gene	This study
**Strains**	**Description**	**Reference**
*E. coli* Xl10-Gold	The strain used for plasmids replication	Stratagene
*K. lactis* GG799	Wild-type yeast	New England Biolabs
*K. lactis* BAP	Yeast containing xl*hasB* and pm*hasA* genes	This study
*K. lactis* BA1	Yeast containing xl*hasB* and hs*hasA1* genes	This study
*K. lactis* BA2	Yeast containing xl*hasB* and hs*hasA2* genes	This study
*K. lactis* BA3	Yeast containing xl*hasB* and hs*hasA3* genes	This study

**Table 2 microorganisms-07-00294-t002:** Final OD_600_ and Growth Rate µ_MAX_ (h^−1^) of all strains constructed in this study. The strains were grown in Shake Flasks. Values were obtained from biological triplicate.

Strain	GG799	BAP	BA1	BA2	BA3
**Final OD_600_**	42 ± 3.5	16.3 ± 5.2	36.9 ± 5.0	37.6 ± 1.2	38.7 ± 3.5
**µ_MAX_ (h^−1^)**	0.33 ± 0.03	0.28 ± 0.03	0.34 ± 0.04	0.35 ± 0.02	0.30 ± 0.03

**Table 3 microorganisms-07-00294-t003:** Final OD_600_, Growth Rate µ_MAX_ (h^−1^), Hyaluronic Acid production (g/L), Hyaluronic Acid Molecular Weight (MDa) and Yields of biomass (x), glycerol (gly), ethanol (et), lactate (lac) and acetate (ace) for all strains constructed. The strains were grown in bioreactor. Values were obtained from biological triplicate.

Strain	Final OD_600_	µ_MAX_ (h^−1^)	HA (g/L)	HA MW (MDa)	Y _x/s_	Y _gly/s_	Y _Et/s_	Y _lac/s_	Y _ace/s_
**GG799**	77.6 ± 1.4	0.30 ± 0.06	0	−	0.42 ± 0.03	0.30 ± 0.01	0.04 ± 0.02	0.02 ± 0.02	0
**BAP**	40.2 ± 5.9	0.31 ± 0.03	1.89 ± 0.2	2.09 ± 0.01	0.34 ± 0.02	0.04 ± 0.02	0.32 ± 0.04	0	0.06 ± 0.01

**Table 4 microorganisms-07-00294-t004:** Engineered microorganisms for HA production, strain construction strategy, HA titers, and molecular weight. Only recombinant strains containing *hasA* and *hasB* genes were considered here.

Microorganism	HasA Source	Promoter	Strain Name	HA (g/L)	Molecular Weight (MDa)	Quantification Method	Substrate Initial Concentration	Ref.
*Enterococcus faecalis* *Escherichia coli*	*S. pyogenes*	--^a^	--^a^	0.69 0.08	--^a^	Carbazole	--^a^	[44]
*Bacillus subtilis*	*S. equisimilis*	P*_amyQ_*-Constitutive	RB184	0.81	1.2	Carbazole	--^b^	[45]
*Lactococcus lactis*	*S. zooepidemicus*	P*_NisA_*-Inductive	LL-NAB	0.65	-- ^a^	HPLC	10 g/L	[46]
*Bacillus subtilis*	*S. zooepidemicus*	P*_VegII_*-Constitutive	RB-AB	0.84	-- ^a^	Carbazole	10 g/L	[47]
*Agrobacterium* sp.	*P. multocida*	Phage T5-Inductive	ATCC31749	0.3	0.7-2	Carbazole	42 g/L	[22]
*Escherichia coli*	*S. equisimilis * *S. pyogenes*	P*_BAD_*-Inductive	sseAB spAB	0.2 0.01	1.9	Carbazole	16 g/L	[46]
*Escherichia coli*	*P. multocida*	Phage T5-Inductive	JM109/pHK	0.55	1.5	Carbazole	45 g/L	[18]
*Lactococcus lactis*	*S. zooepidemicus*	P*_NisA_*-Inductive	NFHA01	0.59	0.88	Radioimmunoassay	20 g/L	[48]
*Lactococcus lactis*	*S. zooepidemicus*	P*_NisA_*-Inductive	SJR2	0.11	2.8	Carbazole	15 g/L	[49]
*Bacillus subtilis*	*P. multocida*	Inductive	--^c^	--^c^	5.43	Carbazole	20 g/L	[20]
*Lactococcus lactis*	*S. zooepidemicus*	P*_NisA_*-Inductive	VRJ2AB	0.14	4.30	Carbazole	10 g/L	[50]
*Pichia pastoris*	*Xenopus laevis*	P*_GAP_*-Constitutive P*_AOX1_*-Inductive	EJ	0.2	0.25	Carbazole	40 g/L	[26]
*Streptomyces albulus*	*S. zooepidemicus*	P*_PLS_*-Inductive	pJHA3	5.1	2	Carbazole	^d^ 60 g/L	[24]
*Bacillus subtilis*	*S. zooepidemicus*	P*_xylA_*-Inductive	pP43-D	2.05	1.76	Carbazole	50 g/L	[19]
*Corynebacterium glutamicum*	*S. equisimilis*	P_SOD_-Constitutive P_dapB_-Constitutive	pXMJ19-PdapB	-- ^a^ 0.14	-- ^a^ -- ^a^	CTAB	40 g/L	[23]
*Corynebacterium glutamicum*	*S. equisimilis*	P*_TAC_*-Inductive	AB	5.4	1.28	CTAB	40 g/L	[51]
*Bacillus subtilis*	*S. equisimilis*	P*_GRAC_*-Inductive	AW008	0.48	1.95	Carbazole	20 g/L	[21]
*Kluyveromyces lactis*	*P. multocida*	P*_LAC4_*-Inductive	BAP	1.89	2.09	Carbazole	40 g/L	This study

^a^. Information not available. ^b^. The initial concentration of substrate during fermentation is not cited. ^c^. No results were presented by study with strains containing only the *hasA* and *hasB* genes. ^d^. The glucose concentration was maintained at 5% by a feed pump.

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
