# Peer review of "Heterologous Hyaluronic Acid Production in Kluyveromyces lactis"

_microorganisms, 2019, doi:10.3390/microorganisms7090294_

Round 1

Reviewer 1 Report

In the manuscript, "Heterologous Hyaluronic Acid production in Kluyveromyces lactis" Gomes et al. develop a heterologous strain of K. lactis capable of producing hyaluronic acid. The authors present the need for alternative sources of HA and present their developed platform as such. Overall, the experimental effort within the text is sound and the development of this HA-producing yeast is of interest to the readership of Microorganisms. However, the manuscript requires significant edits to the text. As written, the authors often obfuscate the intent and scope of their efforts with unclear language that often reads as a first draft. The manuscript also requires more thorough referencing throughout. Below are some specific issues that the authors should address.

pg1;line32: The authors indicate a study and statement from "Grand View Research" but do not provide a reference for the reader to find the included data. Is this organization a reputable body for such data? 

pg1;line35: "use HA for protecting the bones movement, ophthalmologic surgery, and for wrinkle reduction" this line depicts some of the unwieldy language that might confuse readers. The article "the" before bones is not necessary. As written, the line seems to suggest that HA protects bone movement as well as ophthalmologic surgery.

figure 1: Overall the figure appropriately conveys the HA biosynthetic pathway, but a cleaner depiction of the chemical structures more in line with modern methodology would better communicate this route.

paragraph from pg2;line40 to pg3;line53: This paragraph provides an example of the under-referencing throughout the manuscript. The paragraph includes critical exposition about the biosynthesis of HA with detailed descriptions of the classes of HA synthases, but only includes 2 references. Adding references to better direct the readership to the foundational work about HA biosynthesis is critical here and will significantly strengthen the text.

pg3;line89: Again, the sentence beginning with "All the enzymes utilized" specifically states that prior kinetic studies informed the authors' efforts but no references are provided for these studies.

In Table 2 it would be nice to also see the BAP HA titers observed in shake flask experiments.

Throughout the manuscript: The authors should correct the text about chemical formula throughout the text.

Throughout the manuscript: The authors should ensure the same font is used throughout the manuscript.

Author Response

Dear reviewer,

Thank you very much for all your valuable comments. All suggestions certainly helped to enhance the quality of the presented work. Specific answers are detailed bellow. We hope this new version of the manuscript is suitable for publication in Microorganisms journal.

pg1;line32: The authors indicate a study and statement from "Grand View Research" but do not provide a reference for the reader to find the included data. Is this organization a reputable body for such data? 

The “Grand View Research” is a company generally utilized for show the market value of Hyaluronic Acid (HA). Other studies also use the same reference to show about the market, for example, the study “Hyaluronic Acid in the Third Millennium” (https://www.mdpi.com/2073-4360/10/7/701).

Therefore the manuscript has been modified to include this research as a reference.

pg1;line35: "use HA for protecting the bones movement, ophthalmologic surgery, and for wrinkle reduction" this line depicts some of the unwieldy language that might confuse readers. The article "the" before bones is not necessary. As written, the line seems to suggest that HA protects bone movement as well as ophthalmologic surgery.

The sentence has been changed to:

Its increased demand over the years is associated with its application in various fields related to aesthetics, pharmaceuticals, and medicine. Currently, several medical techniques use HA such as in orthopedic [6] and ophthalmologist procedures and for wrinkle reduction in anti-aging treatment[7].

figure 1: Overall the figure appropriately conveys the HA biosynthetic pathway, but a cleaner depiction of the chemical structures more in line with modern methodology would better communicate this route.

We changed the figure as suggested, removing the chemical formula. In fact, the new version is clearer and shorter, however, in our opinion, this change resulted in lost information and it was more poorly represented. Does the reviewer have a punctual suggestion about what should be changed? We can try as many versions until the reviewer is satisfied.

paragraph from pg2;line40 to pg3;line53: This paragraph provides an example of the under-referencing throughout the manuscript. The paragraph includes critical exposition about the biosynthesis of HA with detailed descriptions of the classes of HA synthases, but only includes 2 references. Adding references to better direct the readership to the foundational work about HA biosynthesis is critical here and will significantly strengthen the text.

In the manuscript the following 5 references were added:

Spicer, A.P.; Seldin, M.F.; Olsen, A.S.; Brown, N.; Wells, D.E.; Doggett, N.A.; Itano, N.; Kimata, K.; Inazawa, J.; McDonald, J.A. Chromosomal localization of the human and mouse hyaluronan synthase genes. Genomics 1997, 41, 493–497. DeAngelis, P.L. Hyaluronan synthases: Fascinating glycosyltransferases from vertebrates, bacterial pathogens, and algal viruses. Cell. Mol. Life Sci. 1999, 56, 670–682. DeAngelis, P.L. Molecular directionality of polysaccharide polymerization by the Pasteurella multocida hyaluronan synthase. J. Biol. Chem. 1999, 274, 26557–26562. DeAngelis, P.L. Enzymological characterization of the Pasteurella multocida hyaluronic acid synthase. Biochemistry 1996, 35, 9768–9771. Weigel, P.H.; DeAngelis, P.L. Hyaluronan synthases: A decade-plus of novel glycosyltransferases. J. Biol. Chem. 2007, 282, 36777–36781.

pg3;line89: Again, the sentence beginning with "All the enzymes utilized" specifically states that prior kinetic studies informed the authors' efforts but no references are provided for these studies.

In the manuscript the following 5 references were added:

Itano, N.; Kimata, K. Mammalian Hyaluronan Synthases. IUBMB Life 2002, 1, 195–199. Itano, N.; Sawai, T.; Yoshida, M.; Lenas, P.; Yamada, Y.; Imagawa, M.; Shinomura, T.; Hamaguchi, M.; Yoshida, Y.; Ohnuki, Y.; et al. Three isoforms of mammalian hyaluronan synthases have distinct enzymatic properties. J. Biol. Chem. 1999, 274, 25085–25092. Siiskonen, H.; Oikari, S.; Pasonen-Seppänen, S.; Rilla, K. Hyaluronan synthase 1: A mysterious enzyme with unexpected functions. Front. Immunol. 2015, 6, 1–11. Sussmann, M.; Sarbia, M.; Meyer-Kirchrath, J.; Nüsing, R.M.; Schrör, K.; Fischer, J.W. Induction of Hyaluronic Acid Synthase 2 (HAS2) in Human Vascular Smooth Muscle Cells by Vasodilatory Prostaglandins. Circ. Res. 2004, 94, 592–600. Makkonen, K.M.; Pasonen-Seppänen, S.; Törrönen, K.; Tammi, M.I.; Carlberg, C. Regulation of the hyaluronan synthase 2 gene by convergence in cyclic AMP response element-binding protein and retinoid acid receptor signaling. J. Biol. Chem. 2009, 284, 18270–18281.

In Table 2 it would be nice to also see the BAP HA titers observed in shake flask experiments.

During the experiments in shake flasks, HA was also quantified. Nevertheless  the results are not precise due to the impurities present in the rich YPD medium. Consequently, we decided not to show this data. The Bioreactor results, however, showed small deviations and great reliability. A brief discussion of this issue has also been added in the Discussion section in the manuscript as folow: "The impurities present in the sample react with H2SO4 during hydrolysis changing the Carbazole assay color and giving overestimated results. In this study, the HA purified from growth in shake flasks resulted could not be appropriately determined due to impurities of the rich medium (YPD) (Data not showed). Therefore, HA titer is only reported from culture supernatant using a defined medium. Finally, the protocol for Carbazole method utilized for HA quantification in K. lactis is identical to protocol utilized in most of the 13 studies showed in Table 4, and the Calibration Curve is presented in Supplementary Material Figure S23."

Throughout the manuscript: The authors should correct the text about chemical formula throughout the text.

All chemical formulas were corrected.

Throughout the manuscript: The authors should ensure the same font is used throughout the manuscript.

The entire manuscript was formatted to the Journal's recomended font

Reviewer 2 Report

The authors have described the successfull genetic manipulation of a GRAS yeast Kluyveromyces lactis for the production of hyaluronic acid, which is a highly requested compound at the market. The manuscript is written in a concise way, however, I would suggest a professional English proofreading as many minor language errors distract the reader from the interesting contents of the paper. Together with language proofreading, I propose minor amendments to improve the manuscript:

All over the text, the names of the compounds and enzymes should not be initiated with upper case letters.

l. 93-97 could be omitted, these are results.

l. 207.., the supernatant was filtered..

Please check the fonts as they vary all over the text.

l. 234 and further on: transcription values

l. 268 Transfer Figure S22 from the Supplement to the main text.

Author Response

Dear Reviewer,

Thank you very much for your valuable comments. All the suggestions certainly helped to enhance work quality. Specific comments on the manuscript are answered below. We hope this new version is suitable for publication at Microorganisms Journal.

All over the text, the names of the compounds and enzymes should not be initiated with upper case letters.

Answer: All these terms were reviewed and modified.

93-97 could be omitted; these are results.

Answer: The entire paragraph was removed.

207.., the supernatant was filtered.

Answer: The sentence has been modified.

Please check the fonts as they vary all over the text.

Answer: All the manuscript was formatted with the same font.

234 and further on: transcription values

answer: All these terms were modified.

268 Transfer Figure S22 from the Supplement to the main text.

Answer: Figure S22 is now Figure 4 in the manuscript. We also included the chromatogram of the commercial HA, for comparison.

Round 2

Reviewer 1 Report

The authors have sufficiently addressed the major concerns previously provided. As far as the formatting of Figure 1 is concerned, I suggest that the authors include the figure that they consider to best reflect the biosynthetic pathway especially if the authors consider the revised version to limit the conveyed information. The original figure was by no means incorrect or inappropriate.